# Efficacy of Laser Doppler Flowmetry, as a Diagnostic Tool in Assessing Pulp Vitality of Traumatised Teeth: A Split Mouth Clinical Study

**DOI:** 10.3390/jpm11080801

**Published:** 2021-08-17

**Authors:** Ani Belcheva, Maria Shindova, Reem Hanna

**Affiliations:** 1Department of Paediatric Dentistry, Faculty of Dental Medicine, Medical University-Plovdiv, 15A Vasil Aprilov Blvd, 4002 Plovdiv, Bulgaria; abeltcheva@yahoo.com; 2Department of Surgical Sciences and Integrated Diagnostics, Laser Centre Therapy, University of Genoa, Viale Benedetto XV, 6, 16132 Genoa, Italy; reemhanna@hotmil.com; 3Department of Oral Surgery, Dental Institute, King’s College Hospital NHS Foundation Trust, Denmark Hill, London SE5 9RS, UK

**Keywords:** laser Doppler flowmetry (LDF), pulpal blood flow, pulp vitality, traumatised teeth, paediatric dentistry

## Abstract

Aim: This study aimed to evaluate the efficacy of laser Doppler flowmetry (LDF) in determining the changes in the pulpal blood flow (PBF) during post-traumatic period of the traumatised permanent teeth. Methods: 88 teeth of 44 patients (mean age 10.30 ± 2.38) were recruited according to the eligibility criteria and divided into two groups: test group (44 traumatised teeth) and control group (44 sound and healthy teeth). The measurement of PBF was performed, using a LDF monitor. Results: The analysis of the LDF outcomes in function of diagnosis indicated that the measurements of the traumatised teeth were significantly higher than those of non-traumatised teeth (*p* ˂ 0.05). Conclusions: LDF application provides dentists with fundamental benefits in terms of an early and precise investigation of PBF. In addition, LDF is a useful monitoring tool for revascularization of traumatised teeth and reliable objective diagnostic indicator of pulp vitality. **Trial registration:** ClinicalTrials.gov (Registration number: NCT04967456).

## 1. Introduction

Traumatic injuries of permanent teeth in children and adolescents are very common [1]. Pulp vitality tests play a significant role in achieving an accurate diagnosis and making better treatment plan decision, considering their clinical importance to clinicians [2,3,4]. However, recent studies have reported that the conventional pulp vitality tests were inaccurate and unreliable techniques, assessing the pulp vitality during the post-traumatic period, as well as during the immature teeth examination [5,6,7]. Over the last few years, a direct implementation of the pulpal blood flow (PBF) measurement instrument in dentistry has been achieved [2,3,4,5,6,7,8]. Several authors have reported the use of laser Doppler flowmetry (LDF) to monitor the arterial blood flow of the dental pulp [9,10,11,12,13,14]. As LDF is a reliable and more sensitive technique compared to the conventional pulp vitality tests, Strobl et al. and Emshoff et al. have investigated the PBF of luxated permanent maxillary incisors after tooth splinting technique [12,15,16]. They have reported a valuable diagnostic rate of this device and found a significant decrease in the PBF levels in cases of intrusion injuries. Later, Emshoff et al. studied the relationship between the outcomes of the PBF measurement in subsequent follow-up dental visits and the reported specific values corresponding to the severity of the traumatic injury [17]. In the context, the authors have proposed a predictive modelling that would help clinicians to identify ‘teeth-at-risk’, re-evaluate and reconsider the treatment plan accordingly [17]. Several studies have described the successful use of the LDF for follow-up evaluation after traumatic injury of the permanent teeth and the follow-up protocol was reported [11,12,13,14,15,16]. A review by Moor et al. has examined the use of the LDF in paediatric dentistry, where critical analysis of the advantages and disadvantages of LDF was presented. In addition, authors have discussed areas where LDF use can be improved [13]. In 2019, a cohort study of over 394 recruited teeth, which was conducted by Roeykens et al., has reported that LDF values were highly reliable in observing the differentiation in ‘tooth vitality status’ between the traumatised and non-traumatised teeth [10].

Despite the strong promising results, the implementation of the LDF in modern dentistry and daily clinical practice is scanty. Therefore, our study is aimed to evaluate the efficacy of the LDF in determining the changes in the PBF during the post-traumatic period of traumatised teeth and to establish the potential of LDF usage in daily practice.

## 2. Materials and Methods

### 2.1. Study Design

A split-mouth clinical trial followed the CONSORT guidelines (Figure 1 and Figure 2) [18] and was approved by the Ethics Committee of Medical University of Plovdiv, Bulgaria (Protocol of Approval Code 5/08.07.2021) and registered on a publicly accessible database ClinicalTrials.gov (Registration Number: NCT04967456). The CONSORT checklist (Figure 1) is attached.

#### 2.1.1. Subject Selection

A total of 44 patients (27 male and 17 female) with mean age 10.30 ± 2.38 were screened by the same investigator (A.B.), following the eligibility criteria. The 88 recruited teeth of the 44 patients were divided into two groups: test group—44 traumatised and control group—44 sound and healthy teeth (contralateral homologous tooth with no signs/symptoms of injury).

#### 2.1.2. Eligibility Criteria

##### Inclusion Criteria

Fit and healthy patients aged <18-year-old;Patients with maxillary permanent incisor subjected to any type of traumatic injury (concussion, subluxation, extrusion, lateral luxation, intrusion, crown fracture) without any dentoalveolar trauma involvement (confirmed by a clinical examination and imaging evaluation) and have a contralateral homologous tooth with no signs/symptoms of injury. Treatment of all traumatized teeth was performed, and the established dental trauma guidelines were followed [19];Patients attended their first dental visit within 3 days of the dental trauma;Sensitivity value of the electric pulp test over 200 μA [20] (an electric pulp tester (Scorpion, 405-7A, Optica Laser, Bulgaria) was used to test the vitality of the traumatised and non-traumatised teeth)Patients with at least one vital and non-traumatised maxillary incisor acting as a control tooth;Ability to obtain a verbal acceptance from all the subjects to comply with all the treatments and follow-up timepoints attendance;Ability to obtain written informed consent by the patients’ parents/guardians for treatment, participation in the study and publication.

##### Exclusion Criteria

Patients who were undergoing therapy for neurological conditions or with mental or cognitive problems;Patients who were taking sedative, analgesic, and/or anti-inflammatory medication 7 days prior to the treatment commencement;Patients who have never had any first dental visit or treatment;Patients with systemic diseases or physiological development delay;Patients with active infectious diseases such as; influenza, scarlet fever, etc.Patients with restorations covered more than half the labial surface of the investigated teeth.

### 2.2. Interventions

#### 2.2.1. An Electric Pulp Testing

The measurements of the sensory responses of the traumatised teeth was performed, using an electric pulp tester (Scorpion, 405-7A, Optica Laser, Bulgaria), according to the the following steps:The patient held the passive electrode in the right hand;Instructions were given to the patient-when the slightest irritation in the tooth was felt, he/she must press the passive electrode button;An isolation of the investigated tooth followed;The tooth surfaces were dried with air spray for 15 s;The tip of the active electrode was placed perpendicular to the most sensitive point of the tooth which was at the middle of the incisal edge;The measurement started by pressing the button on the active electrode;The result was read after the button was pressed by the patient.

This electric pulp tester was also used to test the sensitivity of the contralateral homologous tooth of no signs/symptoms of injury.

#### 2.2.2. Laser Doppler Flowmeter (LDF) Description and Utilised Parameters

The measurement of the blood flow of the dental pulp was performed, using a laser Doppler flow monitor “mVMS-LDF2TM” (Moor Instruments Ltd. Millwey Axminster Devon, UK), of 785 nm ± 10 nm emission wavelength with dual-channel laser Doppler monitoring option (Figure 3a). Two probes were used of which each consist of one afferent and one efferent optic fiber, conducting the light to and from the tooth surface. The probe was stainless steel needle-like tube, which is 10–80 mm long and 1.5 mm in diameter. The utilised parameter settings are listed below:Average power—1.0 mV;Diameter of the probe—1.5 mm;Frequency—40 Hz;Monitoring time—180 s (sec).

Prior to each recording session, the LDF was calibrated. The outcomes were measured in arbitrary units (AU).

#### 2.2.3. The Treatment Description

The teeth in both groups; test and control were assessed by LDF, providing an accurate and precise investigation. The temperature in the clinical room was constant and patient’s temperature was measured when they were seated in a “state position” in the dental chair. According to the manufacturer’s recommendations, a patient‘s relaxation of 10 min before the monitoring process was obtained. The measurements of blood flow were performed for 180 s at each session. All the measurements were recorded over a six-year-period (2014–2020) before noon (before 1:00 p.m.).

During the simultaneous measurement of the traumatised and non-traumatised teeth, two identical dental probes were positioned on the vestibular surface of each tooth at approximately 5 mm distance from the gingival margin. In order to ensure an accurate and fixed position of the probes, as well, teeth isolation during the sessions, a custom-made alginate impression covering the incisors and canines was made. The locations of the two probes were determined and the two holes for each probe with a diameter equal to the flowmetry probe (about 1.5 mm) were drilled in the impression (Figure 3). They were inserted into the holes, ensuring a state of a perpendicular position to the enamel surface.

### 2.3. Statistical Analysis

The mean arbitrary unit for each session of the traumatised and control teeth was calculated as an average value by a system data processor. Artifacts due to the movements of the probe or the patient were excluded from this average. The recorded data were saved and stored on a software platform (DTRsoft, Moor instruments Ltd., Millwey Rise Industrial Estate, Axminster EX13 5HU, UK). A paired t-test was used to compare and test the statistically significant differences between the traumatised and non-traumatised teeth. A statistical significance was set at 0.05. For statistical analysis, SPSS version 19.0 (SPSS Inc., Chicago, IL, USA, 2019) was used. In terms of sample size calculation, the lack of fully comparable research and the unknown SD, we conducted a pretest with 20 subjects and considered the values of this subgroup as an estimate. To estimate sample size a *t*-test for paired groups (G* Power software V.3.1 [21] was conducted, since two groups—test group (traumatised teeth) and control group (non-traumatised teeth) were examined. The effect size was determined using the below formula where SD is the pooled SD, an average of the SD of the test and control groups.

$$ES=\frac{Control-Test}{SDpooled}=\frac{2.33-0.33}{3.25}=0.62$$

The error was set at 5% and the power test at 95%. According to the calculation, a sample of 37 patients would be necessary to detect differences in flux values. The anticipated dropout rate for this study was 10%. Adjusting the sample size for this dropout rate resulted in a sample of 41 patients needing to be recruited.

## 3. Results

A total of 88 blood flow records were collected during the measurement session within three days of the traumatic event. The details of these results were described below:

The analysis of LDF outcomes in the function of diagnosis indicated that the measurements of the traumatised teeth were significantly higher than those of the non-traumatised teeth (*p* ˂ 0.05), (Table 1, Figure 4).

In contrast to the measurement of the PBF, using LDF, the results of the electric pulp testing indicated no sensory response of the traumatised teeth after the traumatic event (according to the inclusion criteria, sensitivity value of the electric pulp test over 200 μA).

## 4. Discussion

Our study utilised a LDF device, which is the first dual-channel device, ensuring simultaneous blood flow measurement of both traumatised and non-traumatised teeth with only one session of three-min [9]. Our study has demonstrated that LDF measurement has verified a reliable and precise examination of the blood supply to the pulp of the affected teeth immediately after the traumatic event. In contrast, the current conventional pulp vitality tests require a longer period of time to avoid false-negative results [2,3,4]. On this note, our results have demonstrated the efficacy of LDF in providing effective comparative outcomes between traumatised teeth and those of the homologous contralateral non-traumatised teeth. Hence, this would be a very useful tool for all the dental healthcare workers utilising it in their daily practice to obtain useful information to determine the optimal treatment strategy during the post-traumatic phase and plan the follow-up timepoints. It is noteworthy that the LDF advantageous properties can be considered as a great clinical significance, especially for paediatric dentists, since traumatic injuries are very common in children and adolescents [1].

All the study recruited teeth were unresponsive to the traditional electric pulp vitality test following the described methodology of testing. A study by Kakino et al. showed that the function and vitality of dental pulp were more dependent on the microcirculatory system of the tissue rather than on its sensory nerve response [22]. Due to the false-negative results when immature traumatised teeth were inspected and the patient’s subjective response, the pulp tests that determined the neural response were unsuitable as an assessment tool in paediatric dentistry [2,4,22]. Thus, the innovative methods for pulp vitality testing in paediatric dental patients should be based on the state of the blood supply of the pulp tissue.

In our study, the PBF levels of the traumatised teeth were approximately 25 AU (26.32 ± 17.14), which were similar to the results reported by Setzer et al. and Todea et al. [23,24]. In the period immediately after the trauma, changes in the vascularization of the pulp tissues and its dynamics have occurred due to the injury, resulting in a statistically significant increase in the PBF levels of the affected teeth in the test group compared to those in the control group. In line with this, regular fluctuations of the heart rate on the graphical record oscillations and rhythmic pulsations of the pulpal blood supply were seen in both traumatised and non-traumatised teeth (Figure 5).

As an indication of an active blood supply, these LDF recordings have demonstrated the values of vital teeth investigation (affected and non-affected) with different PBF levels due to the traumatic injury. Studies by Gazelius et al. and Setzer et al. have shown a lack of rhythmic oscillations in non-vital teeth recordings [25]. Interestingly, the results of our study have confirmed that LDF investigation could detect the changes in the PBF levels during the “ischemic phase”, in contrast to the conventional electric or thermal pulp vitality tests [2,3,4,5,6]. Many studies have reported a transient loss of sensitivity of the affected teeth after trauma [1,11,12,15]. In this context, studies conducted by Ingolfsson et al. and Limjeerajarus et al. have shown that the average PBF level of a vital tooth was more than 40% higher than those of a necrotic pulp [9,26].

The possibility for an earlier accurate differentiation of vital and non-vital teeth can be achieved by LDF measurement [9,10,11,12,13,14]. It is of significant clinical importance in providing an adequate and effective dental care for paediatric patients with dental traumatic injuries.

In case of traumatic injury of the immature permanent teeth, the maintenance of pulp vitality and completion of the tooth root development were the foremost and desirable outcomes [1,2,4,7]. Thus, the correct diagnosis and treatment planning immediately after the trauma, as well the follow-up timepoints are crucial for the tooth health prognosis. Thus, LDF monitoring was recommended, as a diagnostic tool along with the clinical examination and radiographic investigations during the post-traumatic phase, in order to postpone or avoid the regular root canal treatment [6,8,10,12].

The results of our study have confirmed the importance of LDF use for diagnosis and measurement of short-term changes in the PBF of traumatised teeth [16]. Additionally, the results of several studies have demonstrated that LDF can be used as a reliable and valid alternative method for pulp vitality test [6,11,12]. However, conducting further studies are necessary to identify the efficiency for diagnosis and measurements of long-term changes in the PBF of traumatised teeth, emanime the correlation between the LDF measurements and the success of the treatment plan, and evaluate the relationship between the PDF values and the histological changes of the pulp tissue [16].

Despite the reliability and accuracy of the LDF, recent studies have emphasised on several limitations of this investigated technology for pulp vitality testing [27,28,29]. In this context, results of studies conducted by Akpinar et al., Gopikrishna et al. and Polat et al. have reported false-positive outcomes due to the interference of laser transmission in case of discoloration of the tooth crown or contamination of the backscattered light from the periodontal tissues [26,27,28]. The following factors have a great impact on the accuracy of the LDF values: the probe design and its position, single or multiple isolated teeth and the mineral contents of the inspected hard dental tissues [14,23,26]. Hence, future studies need to investigate the implementation of a new non-invasive technology, as a diagnostic tool of traumatised teeth.

## 5. Conclusions

Our study has significantly demonstrated the efficacy of LDF application in providing dental practitioners, especially paediatric dentists with fundamental benefits in an early and precise investigation of the short-term changes in the PBF of traumatised teeth. LDF is a useful monitoring tool for detection of the changes in the PBF of the traumatised teeth. Moreover, this technique is a reliable and entirely objective diagnostic indicator of pulp vitality. An extensive research is warranted to enhance further the efficacy of LDF to overcome the technical and environmental factors.

## Figures and Tables

**Figure 1 jpm-11-00801-f001:**
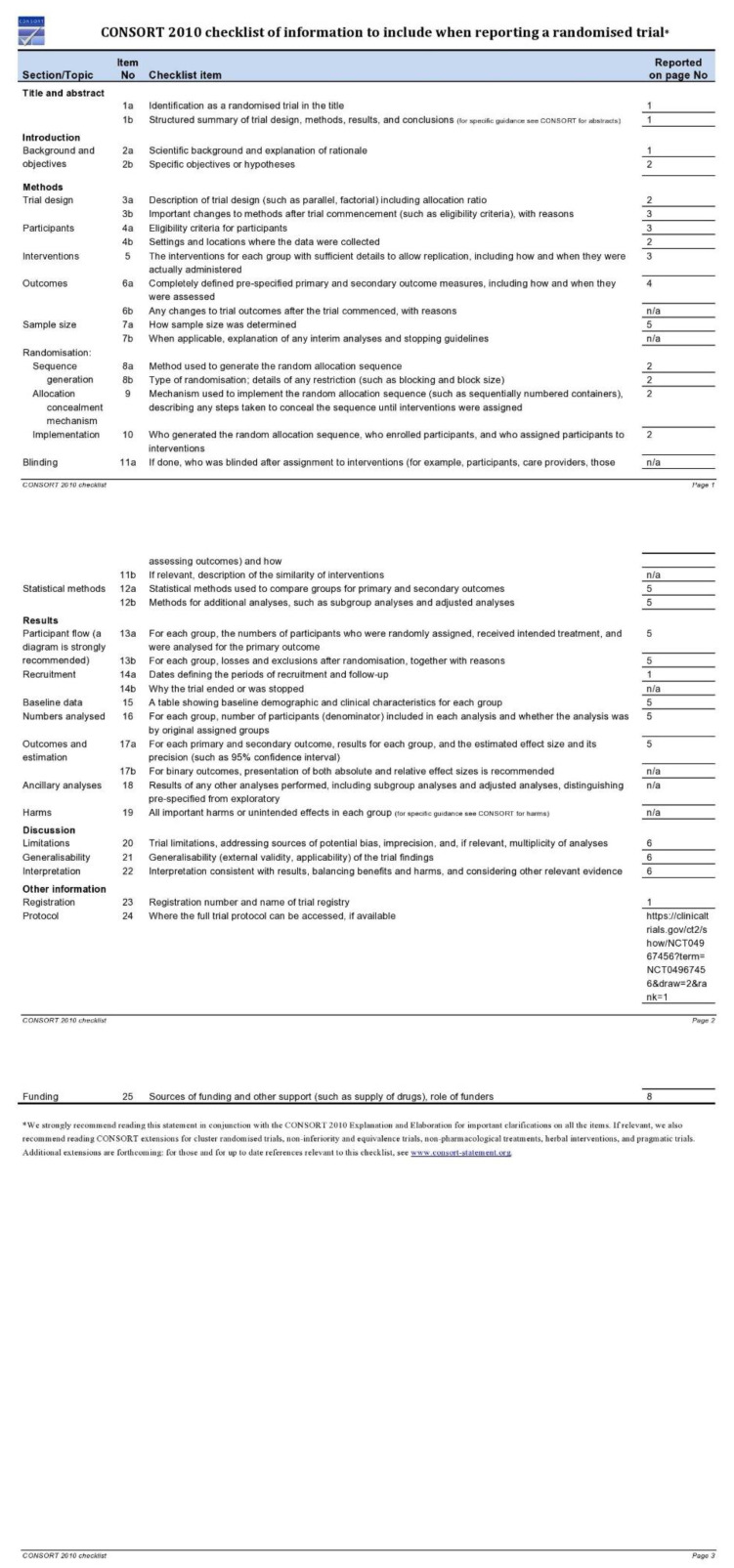
The CONSORT checklist.

**Figure 2 jpm-11-00801-f002:**
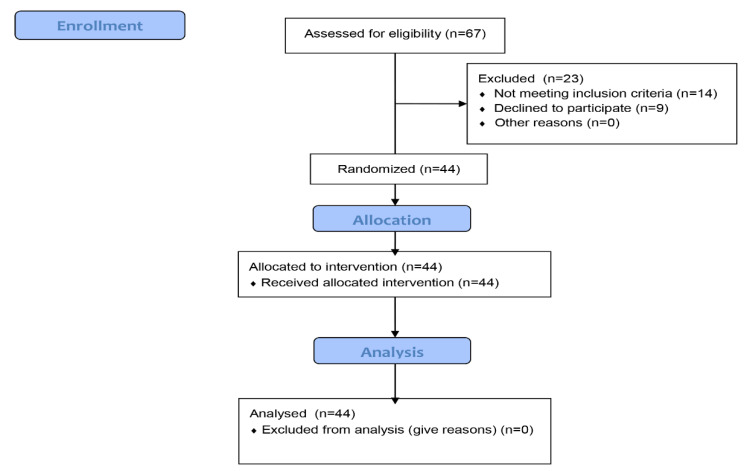
A flow diagram, showing the flow of the participants through each stage of the trial study.

**Figure 3 jpm-11-00801-f003:**
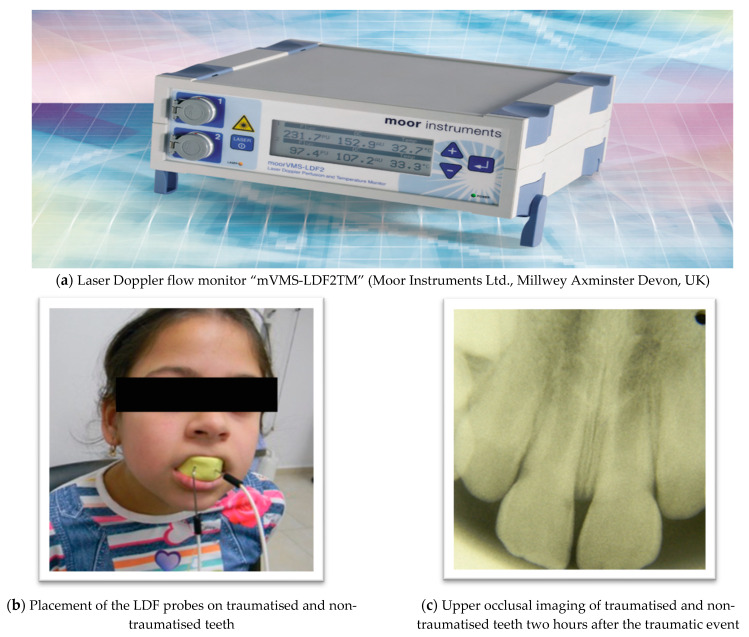
(**a**–**e**) Shows the process of pulpal blood flow measurement.

**Figure 4 jpm-11-00801-f004:**
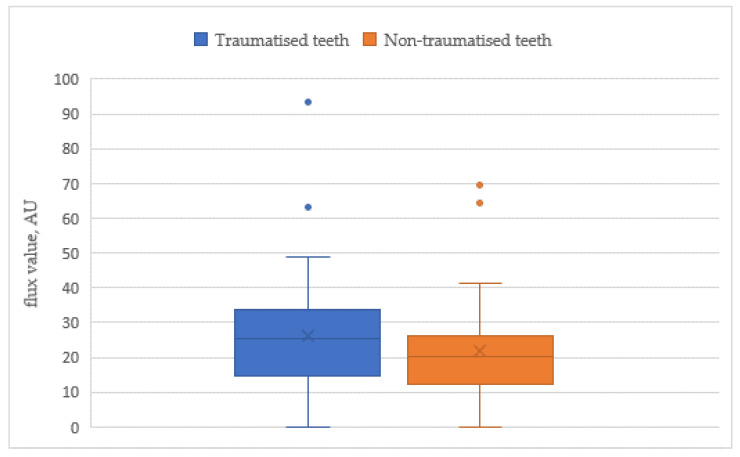
Flux value (AU) of traumatised and non-traumatised teeth.

**Figure 5 jpm-11-00801-f005:**
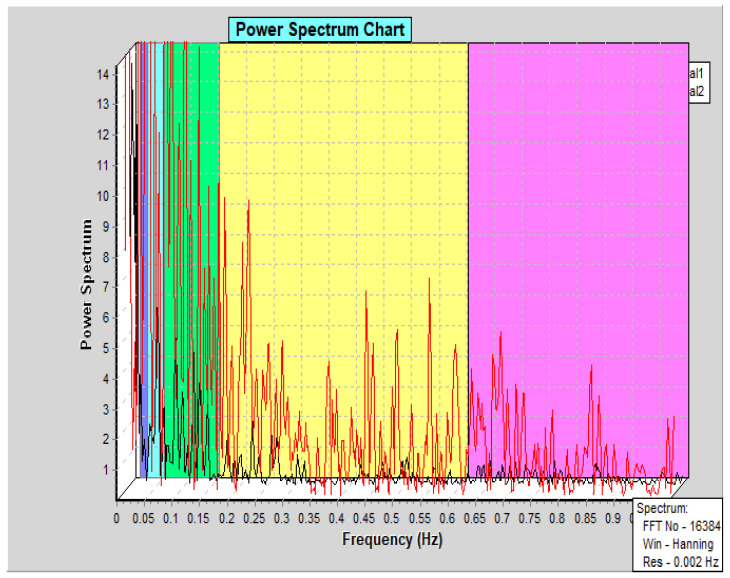
Graphical visualisation of the LDF recording information. A power spectrum density (PSD) graph is displayed. The graph was divided into frequency bands for statistical analysis of the bands. Two parallel measurements of both traumatised and non-traumatised teeth are presented. Each channel was plotted individually (Canal 1, in black, non-traumatised and Canal 2, in red, traumatised teeth).

**Table 1 jpm-11-00801-t001:** Mean and standard deviation of LDF levels of the traumatised and non-traumatised teeth for the first session (N = 44).

Session	Group		*p* Value
Session I	Traumatised teeth	Non-traumatised teeth	
	26.32 ± 17.14 AU	21.53 ± 13.87 AU	0.02

## Data Availability

The data presented in this study are available on request from the corresponding author.

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
