# Peer review of "Efficacy of Laser Doppler Flowmetry, as a Diagnostic Tool in Assessing Pulp Vitality of Traumatised Teeth: A Split Mouth Clinical Study"

_jpm, 2021, doi:10.3390/jpm11080801_

Round 1

Reviewer 1 Report

Figure 3.- says “pdf measurement”

Statistical analysis: please, indicate the statistical power

Please, indicate de Oxygen saturation percentages (%SpO2) by pulp status.

Have you considered to include negative controls? If no, please, explain the reason

Please, explain how the inclusion criteria  “without  any dentoalveolar trauma involvement" was consider.

Have you repeated the measurements? When? Why?

Which is the referenced standard used? Have you compared LDF with other method? Why?

References 20 and 21 are changed. Also, 25.26. Please, review all the references.

Format of references must be reviewed to follow the instructions of the journal.

Author Response

                          7th August 2021

Dear Sir/Madam,

            My co-authors and I appreciate your valuable time in reviewing our manuscript entitledEfficacy of Laser Doppler Flowmetry, as a Diagnostic Tool in Assessing Pulp Vitality of Traumatised teeth. A Split Mouth Clinical Study” under the Special issue Modulating Therapeutic Properties of Oral Tissue-Derived Mesenchymal Stem Cells.

We are very grateful for your comments and suggestions in order to improve our manuscript. We have carefully considered the comments and suggestions and tried our best to address every one of them with evidence-based citation where appropriate. We hope our manuscript after careful revision (attached) meet your high requirements.

            We have provided below with point-by-point responses to your comments and revised version of the manuscript. All modifications in the manuscript have been highlighted in red.

We look forward for your kind consideration of our manuscript for publication.

Thank you.

Yours sincerely,

Maria Shindova, DDS, MSc, PhD

Chief Assistant Professor

Department of Paediatric Dentistry

Faculty of Dental Medicine

Medical University - Plovdiv

3 Hristo Botev Blvd
4000 Plovdiv, Bulgaria

email: mariya.shindova@gmail.com

Figure 3.- says “pdf measurement”

There is a correction in the text.

Statistical analysis: please, indicate the statistical power

The significance level was considered as 95% and power was 80%. We have added this in the text of our manuscript.

Please, indicate de Oxygen saturation percentages (%SpO2) by pulp status.

In our research, we have used the Laser Doppler flowmetry not a pulse oximetry, as a diagnostic tool of assessment. Thus, we did not measure the %SpO2, but the later tool analysed the oxygen saturation as an indicator. Just a note to say that the measurement of the oxygen saturation percentages (%SpO2) can be also performed, using a Pulse oximeter probe.

Have you considered to include negative controls? If no, please, explain the reason

Yes, we have included a control group, which was in our inclusion criteria.

Our study design was split mouth trial. We have recruited subjects with traumatised teeth (experimental group) and have had at least one vital, non-traumatised maxillary incisors on the contralateral side, which was or were homologous tooth/teeth with no signs/symptoms of injury, acted as a control group.

If you were referring  to `negative controls` non-vital teeth due to endodontic treatment, we did not consider this , as it would not be beneficial to assess our variables and outcomes.

Please, explain how the inclusion criteria  “without  any dentoalveolar trauma involvement" was consider.

Sorry about the confusion, We have added the correction in the text.

Have you repeated the measurements? When? Why?

Our research is major, therefore, in the current submitted manuscript we have presented the results of the first part of our project `Efficacy of Laser Doppler Flowmetry as a Diagnostic Tool in Assessing Revascularization of Traumatised Teeth`. After the first measurement, three more assessment sessions will be performed as follows: 3, 6, and 12 months after the traumatic event. The results of the follow-up period will be presented in another manuscript.

Which is the referenced standard used? Have you compared LDF with other method? Why?

As we have mentioned in the Discussion section that we have compared our results with those obtained by Setzer et al. [Setzer, F. C.; Challagulla, P.; Kataoka, S. H. H.;Trope, M. Effect of tooth isolation on laser Doppler readings. Int Endod J 2012, 46, 517–522, doi:10.1111/iej.12019.] and Todea et al. [Todea, C.; Canjau, S.; Miron, M.; Vitez, B.; Noditi, G. Laser Doppler Flowmetry Evaluation of the Microcirculation in Dentistry. Microcirculation Revisited - From Molecules to Clinical Practice,2016, doi:10.5772/64926.]. The authors found about 20 AU (17÷22 AU) flux values immediately after the traumatic event when Laser Doppler flowmetry was used as an additional diagnostic tool.

In our study, all the investigated recruited teeth were unresponsive to the traditional electric pulp vitality test following the described methodology of testing. Therefore, we compared the efficacy of the innovative Laser Doppler technology to the traditional, most commonly used and reliable up to now, electric pulp vitality test.

References 20 and 21 are changed. Also, 25.26. Please, review all the references.

Thank you for this. We have made all the corrections and adjsuted in the text.

Format of references must be reviewed to follow the instructions of the journal.

We have made all the corrections and adjsuted in the text

Reviewer 2 Report

We have no reliability assessment and no sample size calculation. Unless authors could add these pieces of information all the assumptions stated in the discussion and conclusion section are mere speculations and unsupported evidence. 

Author Response

                          7th August 2021

Dear Sir/Madam,

            My co-authors and I appreciate your valuable time in reviewing our manuscript entitledEfficacy of Laser Doppler Flowmetry, as a Diagnostic Tool in Assessing Pulp Vitality of Traumatised teeth. A Split Mouth Clinical Study” under the Special issue Modulating Therapeutic Properties of Oral Tissue-Derived Mesenchymal Stem Cells.

We are very grateful for your comments and suggestions in order to improve our manuscript. We have carefully considered the comments and suggestions and tried our best to address every one of them with evidence-based citation where appropriate. We hope our manuscript after careful revision (attached) meet your high requirements.

            We have provided with point-by-point responses to your comments and revised version of the manuscript. All modifications in the manuscript have been highlighted in red.

We look forward for your kind consideration of our manuscript for publication.

Thank you.

Yours sincerely,

Maria Shindova, DDS, MSc, PhD

Chief Assistant Professor

Department of Paediatric Dentistry

Faculty of Dental Medicine

Medical University - Plovdiv

3 Hristo Botev Blvd
4000 Plovdiv, Bulgaria

email: mariya.shindova@gmail.com

We have no reliability assessment and no sample size calculation. Unless authors could add these pieces of information all the assumptions stated in the discussion and conclusion section are mere speculations and unsupported evidence. 

Thank you for this. We have made all the adjustments in the text.

In terms of the sample size calculation, in order to determine the sample size for each group, a priori power analysis was conducted. The significance level was considered as 95% and power was 80%. The sample size was calculated as 80 teeth (40 teeth per group, i.e. a total number of 40 patients).

After the follow-up period, we will assess the Test-retest reliability in order to evaluate the test for stability over time.

Reviewer 3 Report

The article is well written and scientifically sound. 
Some minor issue should be improved 

Based on findings like these 
Lo Giudice G, Alibrandi A, Lipari F, Lizio A, Lauritano F, Cervino G, et al. The coronal tooth fractures: Preliminary evaluation of a three-year follow-up of the anterior teeth direct fragment reattachment technique without additional preparation. Open Dent J 2017;11(1):266-275.

do you think that this kind of evaluation being more precise in short-term diagnosis of the PBF of traumatised teeth, could be more precise in the long term survival of this teeth ?
Does the author have performed a follow-up in this teeth using the same instrument to do a comparison of the short and long term results?

Author Response

                          7th August 2021

Dear Sir/Madam,

            My co-authors and I appreciate your valuable time in reviewing our manuscript entitledEfficacy of Laser Doppler Flowmetry, as a Diagnostic Tool in Assessing Pulp Vitality of Traumatised teeth. A Split Mouth Clinical Study” under the Special issue Modulating Therapeutic Properties of Oral Tissue-Derived Mesenchymal Stem Cells.

We are very grateful for your comments and suggestions in order to improve our manuscript. We have carefully considered the comments and suggestions and tried our best to address every one of them with evidence-based citation where appropriate. We hope our manuscript after careful revision (attached) meet your high requirements.

            We have provided with point-by-point responses to your comments and a revised version of the manuscript. All modifications in the manuscript have been highlighted in red.

We look forward for your kind consideration of our manuscript for publication.

Thank you.

Yours sincerely,

Maria Shindova, DDS, MSc, PhD

Chief Assistant Professor

Department of Paediatric Dentistry

Faculty of Dental Medicine

Medical University - Plovdiv

3 Hristo Botev Blvd
4000 Plovdiv, Bulgaria

email: mariya.shindova@gmail.com

The article is well written and scientifically sound. 
Some minor issue should be improved 

Based on findings like these 
Lo Giudice G, Alibrandi A, Lipari F, Lizio A, Lauritano F, Cervino G, et al. The coronal tooth fractures: Preliminary evaluation of a three-year follow-up of the anterior teeth direct fragment reattachment technique without additional preparation. Open Dent J 2017;11(1):266-275.

do you think that this kind of evaluation being more precise in short-term diagnosis of the PBF of traumatised teeth, could be more precise in the long term survival of this teeth ?

In the cited clinical trial, the authors have relied  only on the signs observed during the clinical examination. Additionally the sample of the investigated subjects was small and the diagnosis was crown fracture.

  1. In our study we have considered at least two different methods for the examination of the traumatic injured teet, which were theclinical and paraclinical. This is considered the standard of assessment.
  2. The aim of our study was to investigate the efficiency of a new device for paraclinical examination, which can be used in daily practice along with the mandatory clinical examination. Moreover, the application of Laser Doppler for pulp vitality testing is the only device till now that can be utilised to ensure detection of the early changes in the pulpal blood (for short-term diagnosis).
  3. In the cited study, the diagnosis was crown fracture and the observed signs were associated with the changes that corresponded mainly with the fractures. In contrast, our research recruited subjects with different types of traumatic injuries, and these signs can not be used in all cases for diagnosis.
  4. In the paper that was cited by the Reviewer, the number of investigated cases is were small (9 patients) for which it was not sufficient to be considered, as a reliable and more precise method for diagnostic in dental traumatology.

Does the author have performed a follow-up in this teeth using the same instrument to do a comparison of the short and long term results?

Thank you for your valuable question. Our project is  a major one for which it has two parts. Our current submitted t manuscript, where we presented the results of the first part of our project. After the first measurement, three more assessment sessions will be performed as follows: 3, 6 and 12 months after the traumatic event. The results of the follow-up periods will be presented in another manuscript.

Reviewer 4 Report

Line 52: Word missing: …conducted by Roeykens et al. which has reported….

Line 55: shoud it better be written as: Therefore, our study aims to evaluate the efficacy….

Line 80:

I would suggest that the nature of the dental trauma should be better described. Were there only contusions of the teeth with or without loosening or were there also extrusions or intrusions or even luxations and avulsions. This is very important to be able to distinguish between preserved blood circulation and possible revascularization.

Line 84:

The aim was to clearly describe that the traumatized teeth did not show any sensitivity and thus possibly a later root canal treatment was imminent. Apparently, an electrical sensitivity test of the pulp was used as a tool for this purpose. It should be described which device was used and which values the healthy opposing teeth showed in contrast. This is important in order to highlight the value of laser Doppler flowmetry as an additional diagnostic tool. Is there a literature reference for the value of 200 uA?

Line 132: “covering” instead of “covered”?

Line 143:

Unfortunately, you can't see anything on the screenshots of the software used. Perhaps a more comprehensive explanation can be given in the caption of what can be seen in the pictures.

Line 152:

I suspect that the readings were normally distributed so that the paired t-test could be applied.

Line 164:

Is it possible to specify a unit? Later in the discussion the unit is “AU”.

Line 166 to line 168:

If the dental traumas were contusions, the result is plausible in my view. But if, for example, avulsions or extrusions were also examined, then no blood flow in the pulp would be expected, at least directly after the trauma. As described in the discussion (line 212), the blood flow in the pulp would then have to be significantly lower. Therefore, from my point of view, it is important to describe exactly what kind of dental trauma is involved. It would also be interesting to know how long the time was between the trauma and the examination.

Line 184:

See line 84

Line 197:

I suspect that a dot is missing after "group" here.

Line 200:

Is the illustration a patient example? The legend is unfortunately covered by the graph. Can you describe exactly what the red and black curves mean? It would also be helpful if the axes could be explained.

Line 241: Conclusions

I think that LDF is very useful for monitoring the pulp and therefore provides valuable information in addition to the usual thermal and electrical methods. It is difficult to judge from the present study whether revascularizations is taking place, because it is not clear whether the vascularization was really completely interrupted. For this, the type of trauma would have to be clearly defined (line 80). For example, in the case of an avulsion, the tooth is briefly outside the alveolus and thus the vascular bundle is always torn off. After replantation, revascularization of the tooth can occur in immature teeth and LDF would then be a good tool to monitor this. However, a different study design would be needed to test this. In the design of this study, during the diagnostic uncertainty of thermal or electrical testing, it is rather the integrity of the vascular-nerve bundle that is checked.

Author Response

                          7th August 2021

Dear Sir/Madam,

            My co-authors and I appreciate your valuable time in reviewing our manuscript entitledEfficacy of Laser Doppler Flowmetry, as a Diagnostic Tool in Assessing Pulp Vitality of Traumatised teeth. A Split Mouth Clinical Study” under the Special issue Modulating Therapeutic Properties of Oral Tissue-Derived Mesenchymal Stem Cells.

We are very grateful for your comments and suggestions in order to improve our manuscript. We have carefully considered the comments and suggestions and tried our best to address every one of them with evidence-based citation where appropriate. We hope our manuscript after careful revision (attached) meet your high requirements.

            We have provided with point-by-point responses to your comments and revised version of the manuscript. All modifications in the manuscript have been highlighted in red.

We look forward for your kind consideration of our manuscript for publication.

Thank you.

Yours sincerely,

Maria Shindova, DDS, MSc, PhD

Chief Assistant Professor

Department of Paediatric Dentistry

Faculty of Dental Medicine

Medical University - Plovdiv

3 Hristo Botev Blvd
4000 Plovdiv, Bulgaria

email: mariya.shindova@gmail.com

Line 52: Word missing: …conducted by Roeykens et al. which has reported….

Thank you. We have added the word to the text.

Line 55: shoud it better be written as: Therefore, our study aims to evaluate the efficacy….

We have made the correction in the text according to your instructions.

Line 80:

I would suggest that the nature of the dental trauma should be better described. Were there only contusions of the teeth with or without loosening or were there also extrusions or intrusions or even luxations and avulsions. This is very important to be able to distinguish between preserved blood circulation and possible revascularization.

We have taken your valuable suggestion and we have expanded in our desctiption and added to the text.

Line 84:

The aim was to clearly describe that the traumatized teeth did not show any sensitivity and thus possibly a later root canal treatment was imminent. Apparently, an electrical sensitivity test of the pulp was used as a tool for this purpose. It should be described which device was used and which values the healthy opposing teeth showed in contrast. This is important in order to highlight the value of laser Doppler flowmetry as an additional diagnostic tool. Is there a literature reference for the value of 200 uA?

Thank you, we have made the correction based on your valuable points and adjusted in the text.

Line 132: “covering” instead of “covered”?

We have made the correction in the text.

Line 143:

Unfortunately, you can't see anything on the screenshots of the software used. Perhaps a more comprehensive explanation can be given in the caption of what can be seen in the pictures.

The figure shows different display modes as well as different types of report. We have made several corrections in the text associated with the size of the figures and a more detailed description of them.

Round 2

Reviewer 1 Report

Thank you for the changes

Author Response

My co-authors and I appreciate you and the reviewers’ valuable time in reviewing our manuscript.

Reviewer 2 Report

Dear Authors, I appreciate that you try to describe sample size calculation. However, I believe you still failed to give a correct description. You have written: "A statistical significance was set at 0.05. For statistical analysis, SPSS version 19.0 (SPSS Inc., 2019) was used. In terms of a sample size calculation, a priori power analysis was conducted in order to determine the sample size of each group. The significance level was considered as 95% and power was 80%. The sample size was calculated as 80 teeth (40 teeth per group, i.e. a total number of 40 patients)."

You did not explain from where you have taken the data for the sample size calculation nor you explain which software you used to perform the calculation (i.e. the precise bibliographic reference and the data used, otherwise you just could pretend you have done it without actually doping it). Please provide a correct sample size calculation otherwise I will not give my positive opinion on your paper. I may suggest to use GPower software (freeware software). Please choose a specific reference and declare all the data used for your calculation so I check in It. Thank you. Please have your English revised

Author Response

We have made all the corrections and adjusted in the text. Also, we have revised and proofread the entire manuscript.

Reviewer 4 Report

Line 83: It would have been useful to describe the type of trauma in more detail. If the dentoalveolar bone was not injured, the majority of cases were probably contusions. It would also be a good idea to provide information on whether reductions and splinting had to be performed. This would give an indication of how severe the dental injuries were.

Line 201: use a paragraph or perhaps a blank line between the text and the description of the table. It would also be desirable to always use the same font size in the descriptions of the tables and graphics and to set these off a bit from the text.

Line 211: The figure is a new addition. These are obviously the individual measurements of the first session. Unfortunately, the units are missing and the description is quite short. How about better using boxplots?

If using the LDF in cases of diagnostic uncertainty when using conventional vitality testing with cooling or electricity, it would of course also have been nice to know something about the further course of the traumatised teeth. If the blood flow was still present in Session I, did these teeth also show a positive result in the electrical test or the test with cooling in the course of a few weeks? If there was a Session I, was there also perhaps a Session II or a Session III? I assume that the values of traumatised and non-traumatised teeth would have been at the same level again in the course of time. Just then, one could well argue that the extra effort of the LDF compared to a simple vitality test can be justified.

Author Response

My co-authors and I appreciate you and the reviewers’ valuable time in reviewing our manuscript.

Line 83: It would have been useful to describe the type of trauma in more detail. If the dentoalveolar bone was not injured, the majority of cases were probably contusions. It would also be a good idea to provide information on whether reductions and splinting had to be performed. This would give an indication of how severe the dental injuries were.

  Thank you for this. We have made all the corrections and adjusted in the text

Line 201: use a paragraph or perhaps a blank line between the text and the description of the table. It would also be desirable to always use the same font size in the descriptions of the tables and graphics and to set these off a bit from the text.

 Thank you for this. We have made all the corrections and adjusted in the text.

Line 211: The figure is a new addition. These are obviously the individual measurements of the first session. Unfortunately, the units are missing and the description is quite short. How about better using boxplots?

Thank you for this. We have made all the corrections and adjusted in the text.

If using the LDF in cases of diagnostic uncertainty when using conventional vitality testing with cooling or electricity, it would of course also have been nice to know something about the further course of the traumatised teeth. If the blood flow was still present in Session I, did these teeth also show a positive result in the electrical test or the test with cooling in the course of a few weeks? If there was a Session I, was there also perhaps a Session II or a Session III? I assume that the values of traumatised and non-traumatised teeth would have been at the same level again in the course of time. Just then, one could well argue that the extra effort of the LDF compared to a simple vitality test can be justified.

Thank you for your valuable question. Our project is a major one for which it has two parts. Our current submitted manuscript, where we presented the results of the first part of our project. After the first measurement, three more assessment sessions will be performed as follows: 3, 6 and 12 months after the traumatic event. The results of the follow-up periods will be presented in another manuscript.